# Effects of Nutrient Manipulation during Peripartum and Suckling Period on Productivity of Hanwoo Cows and Offspring

**DOI:** 10.3390/ani14182633

**Published:** 2024-09-11

**Authors:** Gi-Hwal Son, Na-Hui Kim, So-Hee Lee, Young-Lae Kim, Jun-Sang Ahn, Min-Ji Kim, Jong-Suh Shin, Byung-Ki Park

**Affiliations:** 1Nonghyup Livestock Research Center, Anseong 17558, Republic of Korea; oscar0202@naver.com; 2Department of Animal Science, Kangwon National University, Chuncheon 24341, Republic of Korea; gom7270@naver.com (N.-H.K.); seesohev@naver.com (S.-H.L.); dudfo123@naver.com (Y.-L.K.); mjkim@kangwon.ac.kr (M.-J.K.); jsshin@kangwon.ac.kr (J.-S.S.); 3Noghyup Feed, Seoul 05398, Republic of Korea; dkswns121@naver.com

**Keywords:** nutrition manipulation, Hanwoo cows, reproductive efficiency, Hanwoo offspring, growth performance

## Abstract

**Simple Summary:**

Nutrient manipulation during the peripartum and suckling periods can significantly impact the productivity of Hanwoo cows and their offspring. This study investigated the effects of feeding nutrient-enriched formula feed compared to moderate formula feed on 183 pregnant Hanwoo cows and their 180 offspring. Results indicated that cows receiving the enriched diet had better body weight recovery postpartum, and their offspring showed higher growth performance. Additionally, the interval to first estrus return and days open were reduced in the nutrient-enriched group. These findings suggest that nutrient-enriched formula feed enhances reproductive efficiency in Hanwoo cows and improves growth performance in their offspring.

**Abstract:**

This study investigated the effects of nutrient manipulation during the peripartum and suckling periods on the productivity of Hanwoo cows and their offspring. A total of 183 pregnant cows and their 180 offspring were randomly assigned to either a control group, fed a formula feed with 13.5% crude protein (CP) and 70.5% total digestible nutrients (TDN), or a treatment group, fed nutrient-enriched formula feed with 18.0% CP and 72.5% TDN. Offspring were similarly divided and fed either 17.0% CP and 69.5% TDN (control) or 21.5% CP and 72.5% TDN (treatment). Results showed that body weight recovery was higher in the treatment group, although wither height, body length, and body condition scores were similar between groups. The treatment group exhibited increased chest girth, reduced intervals for first return to estrus, and shorter days open compared to the control group. Plasma non-esterified fatty acids, albumin, and progesterone concentrations of Hanwoo cows varied between groups at the 3 months before and after calving. Offspring in the treatment group had higher body weight and average daily gain at birth, three and six months of age, with higher dry matter intake. These findings suggest that nutrient-enriched formula feed positively influences the reproductive efficiency of Hanwoo cows and the growth performance of their offspring.

## 1. Introduction

Currently, the long-term fattening system (30–31 months) of Hanwoo cattle in Korea is increasing feed costs for farmers. However, excessive shortening of the breeding period to reduce feed costs may decrease the farmer’s profits due to a decrease in carcass weight and marbling score [1]. Improving growth performance and carcass traits, especially in calves, is essential to achieve this. The late pregnancy period of Hanwoo fetuses is characterized by vigorous fetal development, and the postnatal suckling period is crucial for muscle and fat formation in calves [2]. In addition, due to the genetic improvement of Hanwoo, the birth weight of calves and the improvement in growth ability have increased, and more nutrients are required for pregnant Hanwoo cows, but the nutritional level of cow feed has not gone beyond a relatively constant level. Therefore, enhancing nutrition (total digestible nutrients [TDN] and crude protein [CP]) in pre- and post-calving cow feed, as well as suckling calf feed, is important to improve calf growth performance.

It has been shown that changes in fetal and postnatal or early nutritional and endocrine status can result in permanent alterations in the physiology and metabolism of animals [3]. The results of the nutritional metabolic imprinting experiment on early-growing calves showed that it improved the carcass weight and marbling of calves [4,5,6,7]. However, most studies focused on the nutritional metabolic imprinting during the calf stage. Therefore, this study attempted to confirm the nutritional metabolic imprinting effect during the fetal to calf period by strengthening the nutrition of pregnant cows before parturition, not only during the calf period.

On the other hand, improving the reproductive efficiency of Hanwoo cows is crucial. Recently, profitability per head has been declining due to a decreased reproductive efficiency, particularly the conception rate (72.90% in 2021 → 70.73% in 2022) [8]. The nutritional status of cows significantly impacts their reproductive performance. Energy deficiency, in particular, has been shown to decrease luteinizing hormone function, increasing the number of days in estrus and lowering conception rates [9]. Moreover, inadequate nutrient supply to pregnant cows reduces the live and weaning weights of calves and increases mortality [10]. Thus, the nutritional status of cows before and after calving is closely linked to reproductive efficiency, calf growth, and development. Fortifying nutrition during both the prenatal and lactational periods (maternal cows and calves) is crucial despite limited studies linking these nutritional aspects until recently.

Therefore, this study aims to determine the effects of manipulating nutritional (TDN and CP) content in formula feed for peripartum Hanwoo cows and suckling calves on the reproductive efficiency of Hanwoo cows and the growth characteristics of their offspring.

## 2. Materials and Methods

### 2.1. Animals, Treatments, and Management

The study was conducted across four Hanwoo farms involving 183 pregnant Hanwoo cows (3 months before calving, with a body weight of 521.4 ± 86.1 kg) and their 180 offspring (male calves: 88, female calves: 92, birth weight of 30.9 ± 4.4 kg).

Hanwoo cows were divided into two treatment groups: a control group fed formula feed (15.2% crude protein and 71.0% TDN) + roughage and a treatment group fed nutrient-enhanced formula feed (19.9% crude protein and 72.7% TDN) + roughage. Offspring were assigned to two treatments based on their mother’s treatment: a control group fed formula feed (19.2% crude protein and 69.6% TDN) + roughage and a treatment group fed nutrient-enhanced formula feed (23.5% crude protein and 72.7% TDN) + roughage. Considering the difference in forage according to the ranch environment, the mixed feed and forage amount were adjusted to set CP and TDN at the same level.

Regardless of the treatment groups, formula feed was administered at 2.5–4.0 kg/day and roughage at 4.0–7.0 kg/day on dry matter intake (DMI) standards for peripartum Hanwoo cows relative to body weight, as outlined in the feeding standards for Hanwoo [11]. Formula feed and roughage were provided twice daily (8 a.m. and 6 p.m.). Calves had ad libitum access to formula feed and roughage until 6 months of age. Water was always freely available, and other feeding management procedures followed the practices of the experimental farm. The ingredients and chemical compositions of the experimental diets are presented in Table 1 and Table 2.

### 2.2. Growth Performance

The body condition score (BCS) of cows was assessed on a 5-pointscale with 0.5-unit increments (one indicating emaciation and five indicating obesity). Body measurements, such as body height, body length, and chest girth, were taken using a measuring stick and tape (Newton HT-501A, CAS Korea, Sungnam, Republic of Korea). Body weight was measured using a cattle weight-bridge. Dry matter intake of cows and calves was calculated by measuring the remaining amount of formula feed and roughage before the morning feeding and subtracting it from the amount fed. Feed conversion ratio (FCR) was calculated using average daily gain (ADG) and DMI. Ingredients and chemical composition of the experimental feed were analyzed following AOAC [12] methods, while neutral detergent fiber (NDF) and acid detergent fiber (ADF) were analyzed according to Van Soest et al. [13].

### 2.3. Reproductive Performance

The cow’s return to estrus was monitored through visual observation and an estrus sensor (Healthy Cow 24, Allflex, Netanya, Israel). Artificial insemination was performed during the subsequent estrus period for cows returning within 30 days. Pregnancy was confirmed via palpation and ultrasound diagnostics at 60 to 90 days post-insemination, followed by calculation of services per conception. Calving intensity was categorized as follows: 1 (easy calving), 2 (dystocia but unassisted), and 3 (requiring assistance).

### 2.4. Blood Metabolites and Hormones

Blood samples from Hanwoo cows were collected via the jugular vein 3 months before calving, immediately after calving, and 3 months post-calving, before the morning feed, using an 18-gauge needle and 10 mL vacutainer containing heparin (BD Vacutainer^®^ Sodium Heparin^N^ 158 USP Units, Becton Dickinson Co., Franklin Lakes, NJ, USA). Samples were left at 4 °C for 12 h and then centrifuged at 1250× *g* to collect the plasma for analysis. Analyzed blood parameters included albumin (ALB), alanine aminotransferase (ALT), aspartate aminotransferase (AST), blood urea nitrogen (BUN), calcium (Ca), cholesterol (CHO), gamma-glutamyl transferase (GGT), glucose (GLU), magnesium (Mg), non-esterified fatty acids (NEFA), phosphorus (P), triglycerides (TG), and total protein (TP) using an automatic blood analyzer (Hitachi Ltd., Tokyo, Japan). Blood hormones (progesterone, estrogen, cortisol, Anti-Müllerian Hormone) were analyzed using ELISA Kits (MBS734529, MBS2610579, MBS2608983, MyBioSource, San Diego, CA, USA; EK760227, AFG Scientific, Northbrook, IL, USA).

### 2.5. Statistical Analysis

In each ranch, cows were selected as experimental animals through a normality test considering their body weight, and the number of cows tested in each ranch was 50 in Ranch 1, 50 in Ranch 2, 40 in Ranch 3, and 43 in Ranch 4. To set the ranch variable as a random variable, regional and environmental factors between ranches were excluded as much as possible.

All study results were analyzed using *t*-tests with the least significant difference procedure in the SAS package program (release 9.1.3 version, 2005). Significant differences were considered if *p* < 0.05. Considering the difference in forage according to the ranch environment, the mixed feed and forage amount were adjusted to set CP and TDN at the same level.

## 3. Results

### 3.1. Hanwoo Cows

Table 3 presents the effects of nutrient (TDN and CP) manipulation during the 3 months before and after calving on the growth performance of Hanwoo cows. Body weight recovery after calving was higher in the treatment group than in the control group (*p* < 0.05), while DMI was lower in the treatment group compared to the control group (*p* < 0.05).

Table 4 illustrates the effects of nutrient (TDN and CP) manipulation during the 3 months before and after calving on the body conformation traits of Hanwoo cows. There was no significant effect of TDN and CP manipulation in the formula feed on changes in body height and length before and after calving in Hanwoo cows. In the control group, chest girth decreased at 3 months after calving compared to 3 months before calving, whereas in the treatment group, it increased (*p* < 0.05). Numerically, BCS decreased in the control group at 3 months after calving compared to 3 months before calving, whereas no significant difference was observed in the treatment group.

Table 5 shows the effects of TDN and CP manipulation of formula feed on return to estrus interval, days open, service per conception, and calving intensity in Hanwoo cows before and after calving. The return to estrus interval (1st cycle) and days open (1st, 2nd, and 3rd cycles) after calving were shorter in the treatment group compared to the control group (*p* < 0.05). There was no effect of TDN and CP manipulation on the number of services per conception and calving intensity between the treatment groups.

Table 6 presents the effect of TDN and CP manipulation of formula feed on changes in plasma metabolite concentrations in Hanwoo cows before and after calving. Plasma BUN concentrations were higher in the treatment group than in the control group at 3 months after calving (*p* < 0.05). There was no effect of nutrient manipulation during peripartum on the blood metabolites of Hanwoo cows except BUN.

Table 7 presents the effects of TDN and CP manipulation in formula feed for Hanwoo cows before and after calving on changes in plasma hormone concentrations. Overall, no significant effect was observed on plasma progesterone, estrogen, cortisol, and anti-müllerian hormone concentrations at 3 months before calving.

Immediately after calving, plasma progesterone concentration was lower in the treatment group compared to the control group (*p* < 0.05), whereas differences in plasma estrogen, cortisol, and anti-müllerian hormone concentrations between treatment groups were negligible. At 3 months after calving, plasma progesterone and anti-müllerian hormone concentrations were higher in the treatment group than in the control group (*p* < 0.05), while estrogen concentrations were lower in the treatment group compared to the control group (*p* < 0.05).

### 3.2. Hanwoo Calves

Table 8 illustrates the effect of TDN and CP manipulation in formula feed for Hanwoo cows and their offspring before and after calving on body weight, ADG, feed intake, and FCR of Hanwoo calves. The ADG of calves at birth, 3 months of age (at weaning), and 6 months of age, and DMI from birth to 3 months of age, were higher in the treatment group than in the control group (*p* < 0.05). Roughage intake (birth—3 months of age and birth—6 months of age) was also higher in the treatment group compared to the control group (*p* < 0.05), and DMI (birth—3 months of age) was higher in the treatment group (*p* < 0.05). Although statistical significance was not observed, FCR during the experimental period was numerically lower in the treatment group compared to the control group.

Table 9 presents the effect of nutritional manipulation on the body conformation traits of Hanwoo calves. Although not statistically significant, body height, body length, and chest girth at 3 and 6 months of age were numerically higher in the treatment group than in the control group.

## 4. Discussion

The results of this study indicate that increasing the TDN and CP contents of the formula feed in the treatment group led to higher body weight recovery of cows after calving compared to the control group. These findings align with previous research [14], which demonstrated that cows fed a high-energy diet exhibit better body weight recovery compared to those fed a low-energy diet. Additionally, previous studies suggesting a decrease in DMI with higher energy feed levels [15,16] support our findings of lower DMI in treatments with higher energy content in the formula feed. Therefore, based on these results, increasing the TDN and CP levels of the formula feed is judged to have a positive effect on body weight recovery and feed efficiency improvement in peripartum Hanwoo cows.

BCS is widely used to assess the nutritional status of cows [10], and specifically, BCS during pregnancy is known to affect reproductive efficiency [17]. Meanwhile, Choi et al. [18] suggested that maintaining a BCS around 3 for Hanwoo cows at the end of pregnancy is desirable, as a BCS higher than 3 may lead to increased dystocia, decreased conception rate, and higher incidence of follicular cysts. The results of this study indicate that BCS was maintained around 3 before and after calving regardless of the treatment groups, suggesting no occurrence of issues such as dystocia, decreased conception rate, or follicular cysts.

Interestingly, this study found that feeding a high-nutrient diet resulted in less BCS loss in Hanwoo cows after parturition, contrary to the findings of Janovick et al. [16], who reported increased BCS loss after calving with increased pre-calving feed intake. Therefore, it is concluded that TDN and CP manipulation positively influence body conformation traits and BCS of Hanwoo cows, provided that pre- and post-calving diets are fed in similar amounts.

It has been reported that supplying higher nutrients during the lactation period after calving positively affects reproductive function, such as shortening the return to estrus interval and improving conception rates in cows [11]. BCS during pregnancy is known to affect reproductive efficiency [17], and Carvalho et al. [19] reported decreased conception rates in cows with low BCS. Other studies [11] have shown that severe BCS loss is associated with anovulation and anoestrus in cows, with cows achieving a 100% conception rate at BCS levels of 2.6–3.0 and 3.1–4.0 at insemination, compared to 73.4% conception rate in cows with a BCS of less than 2.5. The results of this study show that no reproductive disorders occurred regardless of the treatment groups, indicating adequate nutrient supply without excess or deficiency. Improved reproductive efficiency in the treatment group compared to the control group is presumed due to higher weight recovery after calving (Table 3) and less BCS reduction (Table 4) in the treatment group.

Plasma NEFA concentration primarily increases when the body experiences insufficient available energy. Elevated plasma NEFA levels have been linked to reproductive disorders in cows [20], often exacerbated by calving stress, which can temporarily elevate NEFA concentrations [21]. Previous research has shown that feeding high-energy diets to pregnant cows results in low NEFA levels before calving and high levels after calving [16], consistent with findings in this study where NEFA concentrations were lower in the treatment group at 3 months before calving but higher at 3 months after calving compared to the control group. However, Farman et al. [22] reported that high NEFA concentrations may adversely affect oocyte maturation, potentially lowering conception rates and hindering early embryo development. While the treatment group in our study showed a tendency toward higher post-calving NEFA levels compared to the control group, it was not deemed sufficient to impact return to estrus interval, days open, or service per conception.

High plasma BUN concentrations in cows have been associated with reduced sperm viability due to uterine acidification, thereby decreasing conception rates [23]. Although our study indicated a slight tendency toward elevated post-calving plasma BUN levels due to TDN and CP adjustments in the formula feed for Hanwoo cows, it was not significant enough to affect conception rates, as evidenced by the shortened return to estrus interval and days open.

ALB, synthesized predominantly in the liver, typically decreases during inflammatory conditions like endometritis and mastitis [24]. While previous studies found no impact of pre-calving energy intake on plasma ALB levels [15], our study observed lower ALB levels in the treatment group at 3 months before calving but higher levels at 3 months after calving compared to the control group, likely due to crude protein manipulation in the formula feed. This aligns with findings by Agenäs et al. [25], suggesting that plasma ALB reflects short-term nutritional status and increases with higher protein intake.

A decrease in plasma CHO concentration in cows can lead to reproductive issues by reducing hormone synthesis and follicle development [26]. However, our study found no significant changes in plasma CHO concentrations across treatment groups, indicating that TDN and CP adjustments in formula feed before and after calving did not impact CHO levels.

Therefore, the results of this study indicate that TDN and CP manipulation in formula feed for Hanwoo cows before and after calving affected the change in protein-related metabolite concentrations (BUN and albumin), but it was considered to have a minimal effect on the concentration of most plasma metabolites.

Progesterone functions to maintain pregnancy after embryo implantation [27]. The higher plasma progesterone concentration observed in the treatment group at 3 months after calving in this study likely reflects enhanced progesterone function. This finding correlates with the shortened days from calving to the 1st days open, as shown in Table 4, possibly due to the shortened estrus return period resulting from TDN and CP manipulation in the formula feed for Hanwoo cows before and after calving.

Estrogen induces estrus through endocrine regulation related to the estrous cycle, with a temporary increase in blood concentration triggering estrus return [28]. In this study, it was observed that plasma estrogen concentration increased 3 months after calving in the treatment group, which exhibited a shorter time to estrus compared to the control group, resulting in lower plasma estrogen concentrations at 3 months after calving.

Anti-müllerian hormone indicates ovarian reserve, with higher concentrations associated with better estrus return and conception rates in cows [29,30]. The higher plasma anti-müllerian hormone concentrations observed in the treatment group at 3 months after calving align with the results shown in Table 5, where the days to estrus return were shortened.

In ruminants, most fetal growth (body weight) occurs during the last 2 months of pregnancy [31], so nutritional management of pregnant cows in the latter half of pregnancy significantly impacts calf birth weight and growth [32]. The results of this study, where nutritional manipulation (TDN and CP) in formula feed for Hanwoo cows before calving increased calf birth weight, align with Gunn et al. [33], who found that nutritional manipulation before calving increased calf birth weight. Furthermore, this study demonstrated higher ADG in calves by month of age in the treatment group due to nutritional manipulation of formula feed for peripartum Hanwoo cows and suckling calves, consistent with previous findings [34].

The results of this study indicated an increase in roughage intake and DMI from birth to 3 months of age (weaning) compared to 3 to 6 months of age, attributed to nutritional manipulation of formula feed for peripartum Hanwoo cows and suckling calves. It is considered that this showed a similar trend to the previous research results [34] that the ADG of calves increased when high-nutrient formula feed was supplied to cows before and after parturition.

The results of this study showed that roughage intake and DMI at birth to 3 months of age (weaning) increased compared to 3 to 6 months of age due to nutritional manipulation of formula feeds for peripartum Hanwoo cows and suckling calves. It is presumed to be due to the difference likely stems from variations in birth weight and ADG increment rates. Therefore, this study suggests that nutritional manipulation of formula feed for peripartum Hanwoo cows and suckling calves significantly influences the growth performance (ADG and DMI) of calves from birth to weaning.

The positive effect of nutritional manipulation in formula feed for peripartum Hanwoo cows and suckling calves on the development of body height, body length, and chest girth in calves aligns with Son’s [35] findings that feeding nutritional enhancement feed to Hanwoo cows in late pregnancy improves calf growth performance.

## 5. Conclusions

It is believed that nutritional manipulation (TDN and CP) of formula feed for Hanwoo cows before and after calving has a positive effect on improving body weight recovery after calving and reproductive efficiency (days to return to estrus and days open). Nutritional manipulation of formula feed for peripartum cows and suckling calves is also beneficial in improving the growth performance of calves.

## Figures and Tables

**Table 1 animals-14-02633-t001:** Chemical composition of experimental diets for Hanwoo cows (DM basis).

Items	Formula Feed		Rouhage ^1^						
	C ^2^	T ^3^	OGS ^4^	ALF ^5^	ARS ^6^	RH ^7^	IRG ^8^	RS ^9^ I	RS II
DM ^10^ (%)	90.62 ± 0.72	91.93 ± 1.28	94.20 ± 0.12	90.17 ± 0.40	94.16 ± 0.40	82.37 ± 0.43	80.60 ± 1.30	86.40 ± 0.95	83.56 ± 1.10
CP ^11^ (%)	15.23 ± 0.43	19.91 ± 0.36	4.99 ± 0.03	19.63 ± 0.22	4.67 ± 0.34	10.76 ± 0.15	6.45 ± 0.16	4.98 ± 0.21	5.74 ± 0.31
CF ^12^ (%)	5.85 ± 1.29	6.20 ± 1.18	36.94 ± 0.54	38.04 ± 0.40	39.40 ± 0.61	47.47 ± 0.32	38.09 ± 0.73	38.77 ± 0.66	38.42 ± 0.88
EE ^13^ (%)	3.53 ± 0.45	2.94 ± 0.53	0.32 ± 0.06	0.44 ± 0.12	0.74 ± 0.21	1.84 ± 1.13	1.12 ± 0.20	0.93 ± 0.50	1.08 ± 0.46
NDF ^14^ (%)	38.51 ± 6.31	35.90 ± 0.39	71.66 ± 0.47	51.68 ± 0.47	72.96 ± 0.88	86.58 ± 0.77	63.15 ± 0.81	71.76 ± 0.79	67.62 ± 0.81
ADF ^15^ (%)	20.97 ± 0.79	17.30 ± 0.86	49.68 ± 0.03	36.38 ± 0.72	50.98 ± 0.71	56.75 ± 0.27	40.32 ± 0.73	54.17 ± 0.61	47.51 ± 0.66
CA ^16^ (%)	6.95 ± 1.55	7.07 ± 1.43	6.26 ± 0.03	10.87 ± 0.12	5.20 ± 0.16	6.27 ± 0.61	6.58 ± 0.22	12.85 ± 0.24	9.81 ± 0.48
TDN ^17^ (%)	70.95	72.69	51.78	62.82	50.83	53.66	63.07	51.70	57.49

^1^ Roughage: OGS and ALF (8:2) at farm I, ARS and RH (5:5) at farm II, IRG and RS I (5:5) at farm III, and RS II at farm IV; ^2^ C: control; ^3^ T: treatment; ^4^ OGS: orchard grass straw; ^5^ ALF: alfalfa; ^6^ ARS: annual ryegrass straw; ^7^ RH: rye hay; ^8^ IRG: italian ryegrass; ^9^ RS: rice straw; ^10^ DM: dry matter; ^11^ CP: crude protein; ^12^ CF: crude fiber; ^13^ EE: ether extract; ^14^ NDF: neutral detergent fiber; ^15^ ADF: acid detergent fiber; ^16^ CA: crude ash; ^17^ TDN: total digestible nutrients.

**Table 2 animals-14-02633-t002:** Chemical composition of experimental diets for Hanwoo calves (DM basis).

Items	Formula Feed		Rouhage ^1^				
	C ^2^	T ^3^	MH I ^4^	ALF ^5^	ARS ^6^	MH II ^7^	TIM ^8^
DM ^9^ (%)	91.40 ± 0.76	91.50 ± 0.53	94.56 ± 0.10	90.08 ± 0.40	94.10 ± 0.40	87.71 ± 0.15	93.60 ± 0.81
CP ^10^ (%)	19.23 ± 0.43	23.53 ± 0.25	6.86 ± 0.07	19.68 ± 0.22	4.68 ± 0.34	7.78 ± 0.36	13.25 ± 0.32
CF ^11^ (%)	12.29 ± 4.11	8.24 ± 3.08	38.06 ± 0.70	38.93 ± 0.40	39.43 ± 0.61	40.48 ± 0.42	43.06 ± 0.55
EE ^12^ (%)	3.12 ± 0.73	4.24 ± 1.43	0.26 ± 0.06	0.48 ± 0.12	0.74 ± 0.21	1.83 ± 0.16	1.50 ± 0.12
NDF ^13^ (%)	34.54 ± 3.74	26.19 ± 6.24	67.12 ± 0.36	51.69 ± 0.47	73.01 ± 0.88	67.37 ± 0.82	65.17 ± 0.93
ADF ^14^ (%)	17.89 ± 1.52	12.91 ± 3.06	48.12 ± 0.73	36.36 ± 0.72	51.01 ± 0.71	57.06 ± 0.19	38.35 ± 0.43
CA ^15^ (%)	8.59 ± 0.86	7.84 ± 1.91	7.41 ± 0.06	10.84 ± 0.12	5.21 ± 0.16	5.46 ± 0.04	6.84 ± 0.11
TDN ^16^ (%)	69.62	72.67	52.87	62.81	50.83	49.10	60.49

^1^ Roughage: MH I and ALF at farm I, ARS at farm II, MH II at farm III, and TIM at farm IV; ^2^ C: control; ^3^ T: treatment; ^4^ MH I: mixed hay I, timothy hay + bluegrass hay; ^5^ ALF: alfalfa; ^6^ ARS: annual ryegrass straw; ^7^ MH II: mixed hay II, timothy hay + bluegrass hay + tall fescue hay; ^8^ TIM: timothy; ^9^ DM: dry matter; ^10^ CP: crude protein; ^11^ CF: crude fiber; ^12^ EE: ether extract; ^13^ NDF: neutral detergent fiber; ^14^ ADF: acid detergent fiber; ^15^ CA: crude ash; ^16^ TDN: total digestible nutrients.

**Table 3 animals-14-02633-t003:** Effects of nutrient manipulation on growth performance of peripartum Hanwoo cows.

Items	Control	Treatment	*p*-Value
Body weight (A) at 3 months before calving (kg)	522.4 *±* 92.6	531.0 *±* 81.9	0.479
Body weight (B) at 3 months after calving (kg)	524.6 *±* 87.0	542.5 *±* 80.3	0.127
Body weight regain (B–A)	2.22 ^b^ *±* 2.21	11.50 ^a^ *±* 1.87	0.001
Formula feed intake (DM kg/d)	2.77 ^a^ *±* 0.22	2.69 ^b^ *±* 0.31	0.027
Roughage intake (DM kg/d)	5.99 *±* 1.05	5.71 *±* 0.94	0.056
Dry matter intake (DM kg/d)	8.77 ^a^ *±* 1.00	8.40 ^b^ *±* 0.93	0.012

^a,b^ Means in the same row with different superscripts differ significantly (*p* < 0.05).

**Table 4 animals-14-02633-t004:** Effects of nutrient manipulation on body conformation traits of peripartum Hanwoo cows.

Items		Control	Treatment	*p*-Value
3 months before calving (A)	Wither height (cm)	128.2 ± 9.4	129.1 ± 5.8	0.391
	Body length (cm)	150.0 ± 9.6	151.4 ± 9.5	0.287
	Chest girth (cm)	191.8 ± 14.7	193.1 ± 12.9	0.530
	BCS ^1^	3.05 ± 0.23	3.02 ± 0.26	0.366
3 months after calving (B)	Wither height (cm)	129.6 ± 9.3	130.5 ± 5.6	0.406
	Body length (cm)	151.7 ± 9.4	153.1 ± 9.3	0.275
	Chest girth (cm)	191.4 ± 13.7	194.6 ± 12.6	0.087
	BCS	3.03 ± 0.34	3.02 ± 0.34	0.957
Difference (B–A)	Wither height (cm)	1.42 ± 1.61	1.39 ± 1.83	0.897
	Body length (cm)	1.64 ± 1.99	1.67 ± 1.95	0.916
	Chest girth (cm)	−0.26 ^b^ ± 3.24	1.36 ^a^ ± 2.94	0.001
	BCS	−0.02 ± 0.35	0.02 ± 0.29	0.329

^a,b^ Means in the same row with different superscripts differ significantly (*p* < 0.05). ^1^ Body condition score: 1 indicating emaciation and 5 indicating obesity.

**Table 5 animals-14-02633-t005:** Changes in reproductive performance of Hanwoo cows at 3 months before and after calving.

Items		Control	Treatment	*p*-Value
Return to estrus interval (days)		41.87 ^a^ ± 15.67	35.75 ^b^ ± 9.46	0.002
Number of days for service	1st	57.60 ^a^ ± 19.79	48.21 ^b^ ± 16.11	0.002
	2nd	85.56 ^a^ ± 24.16	72.07 ^b^ ± 17.71	0.023
	3rd	108.8 ^a^ ± 16.0	86.80 ^b^ ± 6.30	0.025
Service per conception		1.48 ± 0.64	1.38 ± 0.56	0.335
Calving intensity ^1^		1.30 ± 0.60	1.29 ± 0.55	0.889

^a,b^ Means in the same row with different superscripts differ significantly (*p* < 0.05). ^1^ calving intensity: 1 (easy calving), 2 (dystocia but unassisted), 3 (manpower input).

**Table 6 animals-14-02633-t006:** Changes in plasma metabolite concentrations of Hanwoo cows at 3 months before and after calving.

Items	3 Months before Calving			3 Months after Calving		
	Control	Treatment	*p*-Value	Control	Treatment	*p*-Value
GLU ^1^ (mg/dL)	60.62 ± 7.62	60.24 ± 7.96	0.804	59.94 ± 10.27	60.94 ± 11.44	0.606
NEFA ^2^ (uEq/L)	286.3 ± 95.1	262.4 ± 96.6	0.185	297.8 ± 136.7	334.3 ± 150.5	0.157
BUN ^3^ (mg/dL)	9.91 ± 3.55	10.14 ± 2.95	0.723	9.31 ^b^ ± 3.38	11.28 ^a^ ± 3.20	0.001
ALB ^4^ (g/dL)	3.29 ^a^ ± 0.22	3.18 ^b^ ± 0.26	0.016	3.26 ± 0.32	3.28 ± 0.30	0.760
TP ^5^ (g/dL)	6.77 ± 0.73	6.52 ± 0.70	0.063	7.03 ± 0.81	6.99 ± 0.81	0.779
CREA ^6^ (mg/dL)	1.62 ± 0.18	1.62 ± 0.21	0.906	1.47 ± 0.21	1.46 ± 0.23	0.859
CHO ^7^ (mg/dL)	101.6 ± 20.8	95.63 ± 19.09	0.115	111.4 ± 26.4	110.0 ± 25.9	0.753
TG ^8^ (mg/dL)	20.66 ± 5.74	20.49 ± 4.14	0.861	16.19 ± 5.14	14.98 ± 4.74	0.181
AST ^9^ (U/L)	45.98 ± 9.69	44.29 ± 9.71	0.355	53.77 ± 12.80	52.05 ± 12.49	0.454
ALT ^10^ (U/L)	13.02 ± 3.50	12.20 ± 3.63	0.218	13.73 ± 5.80	12.02 ± 4.54	0.068
GGT ^11^ (U/L)	19.73 ± 4.02	18.95 ± 3.42	0.268	21.40 ± 5.65	21.46 ± 4.94	0.957
IP ^12^ (mg/dL)	4.95 ± 0.59	5.07 ± 0.83	0.361	4.90 ± 0.73	4.93 ± 0.67	0.811
Ca ^13^ (mg/dL)	7.71 ± 0.77	7.47 ± 0.75	0.092	7.93 ± 0.65	7.86 ± 0.58	0.534
Mg ^14^ (mg/dL)	1.80 ± 0.18	1.77 ± 0.17	0.412	1.83 ± 0.24	1.85 ± 0.25	0.575

^a,b^ Means in the same row with different superscripts differ significantly (*p* < 0.05). ^1^ GLU: glucose; ^2^ NEFA: non-esterified fatty acid; ^3^ BUN: blood urea nitrogen; ^4^ ALB: albumin; ^5^ TP: total protein; ^6^ CREA: creatinine; ^7^ CHO: cholesterol; ^8^ TG: triglyceride; ^9^ AST: aspartate-amino-transferase; ^10^ ALT: alanine transaminase; ^11^ GGT: γ-glutamyl-transferase; ^12^ IP: inorganic phosphorous; ^13^ Ca: calcium; ^14^ Mg: magnesium.

**Table 7 animals-14-02633-t007:** Changes in plasma hormone concentrations of Hanwoo cows at 3 months before and after calving.

Items		3 Months before Calving	Immediately after Calving	3 Months after Calving
Progesterone (ng/mL)	Control	4.82 ± 1.53	3.54 ^a^ ± 1.61	4.41 ^b^ ± 2.80
	Treatment	4.83 ± 1.63	2.70 ^b^ ± 1.22	6.02 ^a^ ± 3.12
	*p*-value	0.467	0.002	0.018
Estrogen (ng/mL)	Control	12.90 ± 6.84	15.93 ± 8.14	9.93 ^a^ ± 8.31
	Treatment	14.21 ± 6.04	14.20 ± 6.22	6.12 ^b^ ± 5.00
	*p*-value	0.392	0.172	0.017
Cortisol (ng/mL)	Control	4.90 ± 4.01	6.03 ± 3.22	4.14 ± 2.52
	Treatment	5.04 ± 3.63	5.13 ± 2.90	4.83 ± 3.12
	*p*-value	0.610	0.072	0.241
Anti-Müllerian Hormone (ng/mL)	Control	0.26 ± 0.13	0.13 ± 0.02	0.24 ^b^ ± 0.16
	Treatment	0.23 ± 0.13	0.13 ± 0.02	0.37 ^a^ ± 0.11
	*p*-value	0.239	0.701	0.001

^a,b^ Means in the same column with different superscripts differ significantly (*p* < 0.05).

**Table 8 animals-14-02633-t008:** Changes in growth performance of offspring of Hanwoo cows.

Items	Control	Treatment	*p*-Value
Birth to 3 months of age			
Body weight at birth (kg)	29.72 ^b^ ± 4.17	31.89 ^a^ ± 4.30	0.001
Body weight at 3 months of age (kg)	101.6 ^b^ ± 9.2	109.3 ^a^ ± 10.1	0.001
Average daily gain (kg/d)	0.80 ^b^ ± 0.09	0.86 ^a^ ± 0.10	0.001
Intake (DM kg/d)			
Formula feed	0.63 ± 0.05	0.62 ± 0.04	0.754
Roughage	0.19 ^b^ ± 0.03	0.23 ^a^ ± 0.05	0.001
Dry matter	0.80 ^b^ ± 0.07	0.84 ^a^ ± 0.09	0.007
Feed conversion ratio	1.02 ± 0.14	0.99 ± 0.15	0.165
3 to 6 months of age			
Body weight at 3 months of age (kg)	101.6 ^b^ ± 9.2	109.3 ^a^ ± 10.1	0.001
Body weight at 6 months of age (kg)	185.1 ^b^ ± 18.5	198.8 ^a^ ± 18.4	0.001
Average daily gain (kg/d)	0.93 ^b^ ± 0.13	0.99 ^a^ ± 0.11	0.001
Intake (DM kg/d)			
Formula feed	2.51 ± 0.58	2.61 ± 0.64	0.283
Roughage	0.66 ± 0.18	0.68 ± 0.17	0.358
Dry matter	3.16 ± 0.75	3.29 ± 0.80	0.292
Feed conversion ratio	3.47 ± 0.95	3.33 ± 0.83	0.307
Entire period			
Body weight at birth (kg)	29.72 ^b^ ± 4.17	31.89 ^a^ ± 4.30	0.001
Body weight at 6 months of age (kg)	185.1 ^b^ ± 18.5	198.8 ^a^ ± 18.4	0.001
Average daily gain (kg/d))	0.86 ^b^ ± 0.10	0.93 ^a^ ± 0.10	0.001
Intake (DM kg/d)			
Formula feed	1.57 ± 0.30	1.62 ± 0.32	0.306
Roughage	0.45 ^b^ ± 0.12	0.48 ^a^ ± 0.10	0.019
Dry matter	2.02 ± 0.36	2.10 ± 0.37	0.152
Feed conversion ratio	2.32 ± 0.49	2.24 ± 0.44	0.251

^a,b^ Means in the same row with different superscripts differ significantly (*p* < 0.05).

**Table 9 animals-14-02633-t009:** Changes in body conformation traits of offspring of Hanwoo cows.

Items		Control	Treatment	*p*-Value
3 months of age (A)	Wither height (cm)	85.89 ± 5.47	87.56 ± 6.09	0.053
	Body length (cm)	82.12 ± 8.18	84.16 ± 8.94	0.109
	Chest girth (cm)	100.7 ± 8.6	102.5 ± 10.2	0.187
6 months of age (B)	Wither height (cm)	101.1 ± 10.0	101.3 ± 7.3	0.871
	Body length (cm)	102.5 ± 11.6	104.6 ± 13.1	0.272
	Chest girth (cm)	127.4 ± 12.5	129.2 ± 13.7	0.370
Difference (B–A)	Wither height (cm)	15.34 ± 9.05	13.89 ± 4.80	0.183
	Body length (cm)	20.78 ± 8.32	20.83 ± 8.89	0.969
	Chest girth (cm)	26.39 ± 8.25	26.74 ± 8.32	0.785

## Data Availability

The original contributions presented in the study are included in the article.

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
