# Peer review of "Effects of Nutrient Manipulation during Peripartum and Suckling Period on Productivity of Hanwoo Cows and Offspring"

_animals, 2024, doi:10.3390/ani14182633_

Round 1
Reviewer 1 Report
Comments and Suggestions for Authors
The manuscript well prepared and is in good presentation and writing. I only have few comments:
Abstract:
Please make it clear for this sentence: “Plasma non-esterified fatty acids, albumin, and progesterone concentrations varied between groups, indicating metabolic differences”. How different were they and the provide the indication/ implications.
Line 75: in your formula, it was mentioned that CP content is 19.91%, not 18%. Please clarify or modify! Same for the calves, not 21.5 but 23% according to table 2.
To better understand the diet composition, authors need to provide the overall chemical composition of roughage (in addition to the individual composition) and the feed formula or concentrate chemical composition.
Stats: are the data normally distributed?
Conclusion: line 332 -> remove “therefore” in the first paragraph.
Author Response
Comment 1: (Abstract) Please make it clear for this sentence: “Plasma non-esterified fatty acids, albumin, and progesterone concentrations varied between groups, indicating metabolic differences”. How different were they and the provide the indication/ implications.
Responce 1: As it is an abstract, I could not add detailed explanations of metabolic-related items. Therefore, I modified the sentence to "Plasma non-esterified fatty acids, albumin, and progesterone concentrations of Hanwoo cows varied between groups at the 3 months before and after calving." and described the details in the discussion section.
Comment 2: (Line 75) in your formula, it was mentioned that CP content is 19.91%, not 18%. Please clarify or modify! Same for the calves, not 21.5 but 23% according to table 2.
Responce 2: It seems that there was a misunderstanding because the method of presenting the as-fed basis and dry matter was different. Based on the reviewer's comments, the materials and method section was changed to match the values ​​in the table.
Comment 3: To better understand the diet composition, authors need to provide the overall chemical composition of roughage (in addition to the individual composition) and the feed formula or concentrate chemical composition.
Response 3: Although we measured the weight of forage at a specific ratio and fed it to each farm, we do not have the results of the analysis of the mixed forage. Therefore, rather than presenting the analysis of the mixed forage by farm as you commented, we will provide the ratio of forage used by each farm in the footnotes.
Comment 4: (Stats) are the data normally distributed?
Response 4: yes, we checked normality test of data
Comment 5: (Conclusion, line 332) remove “therefore” in the first paragraph.
Response 5: we removed "therefore" according to reviewer's comment.
Reviewer 2 Report
Comments and Suggestions for Authors
Dear authors:
After carefully reviewing your document "Effects of nutrient manipulation during peripartum and suckling period on productivity of Hanwoo cows and offspring" I am attaching a PDF file in which I make some observations and recommendations that may help improve some aspects of the document. I hope you analyze them and discuss the option of accepting them.
Greetings.

Author Response
Comment 1: In lines 42 to 46 the authors write “Shortening the fattening period of Hanwoo cattle is
likely to decrease farmers' profits due to reduced carcass weight and marbling score [1].
Improving growth performance, especially in calves, is essential to achieve this.” This
statement should be clarified, since it can be understood that it would reduce the profits
of producers. I suggest clarifying that it would shorten the fattening period.
Response 1: we have revised that sentence according to reviewer's comment.
Comment 2: In line 70, the authors state that the study involved cows from 4 different farms, however,
they do not mention how many cows in the study were from each of the farms. On the
other hand, the statistical analysis did not consider whether the farm variable influenced
the variables studied, especially considering that, as they clarify in Table 1, there were
differences by farm in the ingredients of the diets. I suggest that the authors clarify how
many cows from each farm were used in the study and whether there were differences in
weight and body conformation between the cows from each farm, as well as including the
farm as an independent variable.
Response 2: The number of experimental animals per farm was presented based on the reviewer's comments. In addition, the content on selecting experimental animals through a normality test according to test weight was added. The content on variable processing by farm was also added to the statistics section.
Comment 3: In line 127 of the materials and methods chapter, the authors indicate that Blood
hormones were analyzed using ELISA Kits, however they do not mention which hormones
were measured. They should include this information.
Response 3: we have added parameter of hormones using ELISA kits
Comment 4: In lines 174 and 175, the authors state that in "this study found that feeding a high nutrient diet resulted in less BCS loss in Hanwoo cows after parturition", however, this is
not observed in the results table (Table 4), since no statistical difference was found for this
variable between the animals subjected to the two diets in the two measurement periods
reported nor in the difference between them.
Response 4: we have revised according to reviewer's comment.
Comment 5: In lines 184 to 186, it is stated that "Although statistical significance was not observed,
there was a numerical decrease in the number of services per conception in the treatment
group compared to the control group." It is important to emphasize that the only
differences that should be reported are the statistical ones, since "numerical differences"
are not relevant because they are not statistically relevant.
Response 5: we have deleted that sentences according to reviewer's comment.
Comment 6: In Table 5, authors are encouraged to include a column title for the cycles (1st, 2nd, and
3rd) and to clarify, both in the text preceding the table and in the table itself, whether they
are referring to estrous cycles. It is also advisable to clarify that the variable "Return to
estrous interval" is reported in days.
Response 6: we have revised table 5 according to reviewer's comment
Comment 7: In Table 5, the lines for cycles 2° and 3° with respect to the return to estrus interval
variable can be omitted (because they are unnecessary), and the date of the first cycle
after delivery can be considered as the return to the cycle.
Response 7: we have deleted 2nd and 3rd days for return to estrus interval.
Comment 8: Also for Table 5 it is important to first describe how they define open days, since they
indicate open days for each cycle (1st, 2nd and 3rd), however, it is considered that open
days is the calving-to-conception interval, that is, the period between parturition and the
following conception of a dairy cow. Therefore, I consider that there should only be one
line and not three for this variable, unless the authors explain why they describe it that
way.
Response 8: Days open does not mean the period from birth to conception, but means days open for service. Therefore, I think it is necessary to provide information on the 1st, 2nd, and 3rd service days. I have revised the terminology for this.
Comment 9: In lines 205 to 209, it is stated that "Plasma NEFA, ALB (p<0.05), and CHO concentrations in Hanwoo cows were numerically lower in the treatment group than in the control group at
3 months before calving. Plasma NEFA and BUN concentrations (p<0.05) tended to be
higher in the treatment group than in the control group at 3 months after calving."
However, when analyzing Table 6, the only variables with differences between the
experimental groups were ALB (3 months before calving) and BUN (3 months after
calving). The other variables showed no differences. It is advisable not to talk about
"trends" or "numerical differences".
Response 9: we have deleted that sentences according to reviewer's comment.
Comment 10: In lines 222 to 225, the authors state that in their study NEFA concentrations were lower in the treatment group at 3 months before calving but higher at 3 months after calving
compared to the control group, however, these differences are not observed in Table 6.
The same situation is found in lines 239 and 240, where the authors state that ALB showed
higher levels at 3 months after calving compared to the control group.
Response 10: we have deleted that sentences according to reviewer's comment.
Comment 11: In line 259 when authors point out that "Means in the same row with different
superscripts differ significantly" should be changed row by column.
Response 11: we have revised according to reviewer's comment.
Comment 12: In lines 276 I suggest that the term "increased earlier" be replaced by increased 3 months after calving.
Response 12: we have revised according to reviewer's comment.
Comment 13: In lines 300 to 302 the authors state that the results of the study, "where nutritional
manipulation (TDN and CP) in formula feed for Hanwoo cows before and after calving
increased calf birth weight." I suggest omitting the "after calving" because, since it is a
variable of birth weight, obviously the feeding of the cows and the feeding of the calves
after this period cannot affect the weight with which they were born,
Response 13: we have deleted "and after" according to reviewer's comment.
Comment 14: In line 307 the authors note that "The results of this study indicated a numerical increase in roughage intake". I suggest removing the word numerical.
Response 14: we have deleted "numerical" according to reviewer's comment.
Comment 15: In line 307 the authors state that "The results of this study indicated a numerical increase in roughage intake". I suggest deleting the word numerical. Likewise, I suggest deleting the
term "numerically increased" from line 314.
Response 15: we have deleted "numerical" according to reviewer's comment.
Comment 16: In lines 317 to 319 the authors state that their results suggest that nutritional
manipulation of formula feed for peripartum Hanwoo cows and suckling calves
significantly influences the growth performance of calves from birth to weaning. In my
opinion, this study shows that the diet of the cows in the Treatment group resulted in
calves born with greater weight than calves of cows in the Control group (29.72b±4.17
31.89a±4.30 respectively, a difference of 7.3%). However, I do not see how the different
diets of cows and calves affected calf weight from calving to weaning, as the weight
differential was maintained between both groups from birth to 3 months of age
(101.6b±9.2 109.3a±10.1; weight differential 7.6%) and from 3 months of age to 6 months
of age (85.1b±18.5 198.8a±18.4; weight differential 7.4%).
Comment 17: Because of the above, what is mentioned in lines 335 to 336 in the conclusions section
"Nutritional manipulation of formula feed for peripartum cows and suckling calves is also
beneficial in improving the growth performance of calves" is not necessarily true with
respect to diet consumption by calves.
Response 16-17: I also think the author's opinion is reasonable. I didn't think to calculate the difference in calf weight by treatment as a %. However, I think the difference in birth weight (2.17 kg), weaning weight (7.7 kg), and 6-month weight (13.7 kg) is due to the nutritional manipulation effect of the feed for 3-6 month-old calves. If the effect is only due to the difference in birth weight, I don't think the difference in daily weight gain (0.93 vs. 0.99) at 3-6 months of age can be explained.
Reviewer 3 Report
Comments and Suggestions for Authors
This study provides a detailed analysis of the effects of nutrient manipulation during the peripartum and suckling periods on the productivity of Hanwoo cows and their offspring. The experimental design is robust, involving a substantial sample size of 183 pregnant cows and 180 offspring, which were randomly assigned to control and treatment groups.
Specific comments:
The introduction section needs to be strengthened, focusing on previous studies and novelty offered here
It would be better to separate the results and discussion sections.
WHat about the economic impact or costing of the feed?
Author Response
Comment 1: The introduction section needs to be strengthened, focusing on previous studies and novelty offered here
Response 1: In response to the reviewer's comments, we added previous related studies to strengthen the introduction and described the uniqueness of this study.
Comment 2: It would be better to separate the results and discussion sections.
Response 2: We separated the Results and Discussion sections based on reviewer comments.
Comment 3: What about the economic impact or costing of the feed?
Response 3: Of course, as the reviewer expected, the feed cost increased due to nutritional manipulation. However, since the offspring of the cows were not yet slaughtered, an economic analysis of the productivity (income) aspect was not possible. Simply presenting the feed cost alone could be disadvantageous in interpreting the paper, so it was not presented. I ask for the reviewer's generous understanding in this regard.
Round 2
Reviewer 2 Report
Comments and Suggestions for Authors
Thank you for addressing the comments I made about your article.